# ToM-agent: Large Language Models as Theory of Mind Aware Generative Agents with Counterfactual Reflection

## Abstract

Recent studies have increasingly demonstrated that large language models (LLMs) possess significant **theory of mind (ToM)** capabilities, showing the potential for simulating the tracking of mental states in generative agents. In this study, we propose a novel paradigm called **ToM-agent**, designed to empower LLMs-based generative agents to simulate ToM in open-domain conversational interactions. ToM-agent disentangles the confidence from mental states, facilitating the emulation of an agent's perception of its counterpart's mental states, such as **beliefs, desires, and intentions (BDIs)**. Using past conversation history and verbal reflections, ToM-Agent can dynamically adjust counterparts' inferred BDIs, along with related confidence levels. We further put forth a counterfactual intervention method that reflects on the gap between the predicted responses of counterparts and their real utterances, thereby enhancing the efficiency of reflection. Leveraging empathetic and persuasion dialogue datasets, we assess the advantages of implementing the ToM-agent with downstream tasks, as well as its performance in both the *first-order* and the *second-order* ToM. Our findings indicate that the ToM-agent can grasp the underlying reasons for their counterpart's behaviors beyond mere semantic-emotional supporting or decision-making based on common sense, providing new insights for studying large-scale LLMs-based simulation of human social behaviors. *The codes of this project will be made publicly available for camera ready.*

## 1 Introduction

Generative agents (Park et al., 2023; Wang et al., 2023), which are computational interactive agents with critical components such as memory, observation, planning, and reflection, have been proposed to simulate believable human behavior during conversational interactions by fusing with LLMs (Radford et al., 2019; Brown et al., 2020; Ouyang et al., 2022; OpenAI, 2023; Touvron et al., 2023). Nevertheless, the limitations of LLMs in generating extended, coherent dialogues are well-documented, particularly their proclivity for generating hallucinated or inconsistent content (Rawte et al., 2023; Zhang et al., 2023). These shortcomings are especially problematic when the purpose of the conversation extends beyond simple information exchange to include emotive or persuasive elements, such as in scenarios of emotional support, sales, or persuasive communication (Hu et al., 2023; tse Huang et al., 2023; Remountakis et al., 2023). Such situations necessitate not merely the exchange of factual information, but also the articulation of nuanced demands or emotional appeals (Yakura, 2023), which current LLMs architectures struggle to maintain across natural and prolonged conversational sequences (Zheng et al., 2023).

Referring to psychology science, it is noticed that human normally do not only express their emotions or demands during interaction but also care about their own or counterpart's mental status, such as **beliefs, desires, and intentions (BDIs)** has been understood or satisfied during the communications (Dvash & Shamay-Tsoory, 2014; Grazzani et al., 2018; Rusch et al., 2020). **Theory of mind (ToM)**, a cognitive skill that enables an individual to track the BDIs, emotions, and knowledge of others, plays a crucial role in effective communication, self-consciousness, empathetic emotional support, and decision-making when human beings interact with each other, as well as interactions between human beings and artificial intelligence (AI) (Georgeff et al., 1999; Rabinowitz et al., 2018;

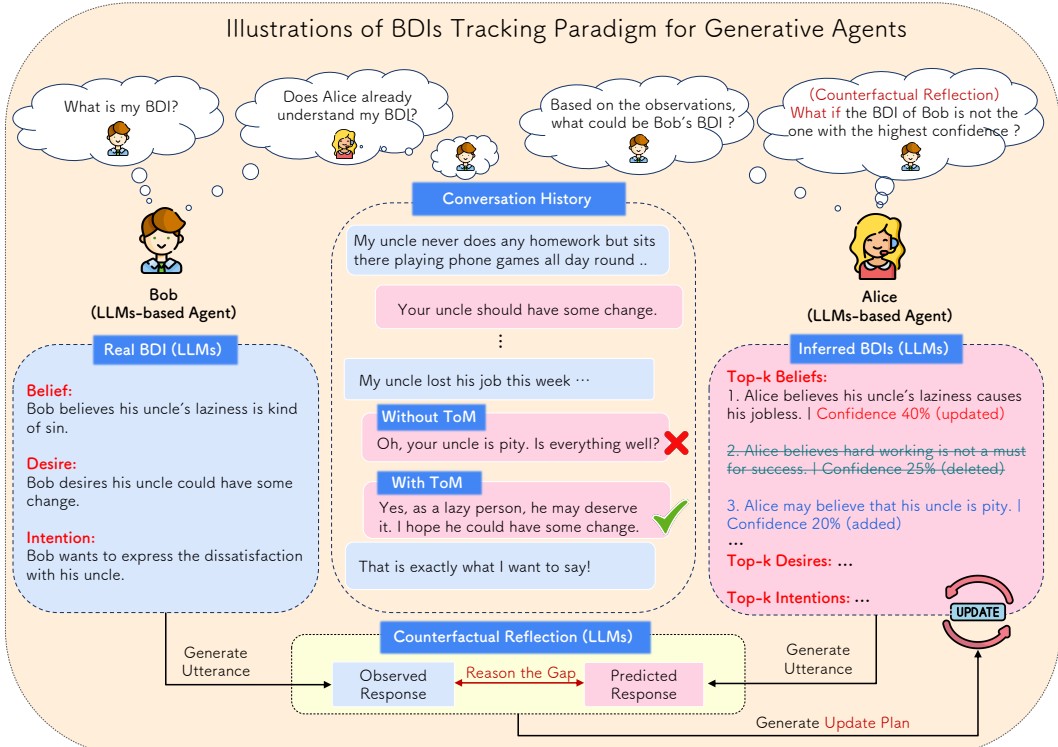

Figure 1: Illustrations of proposed ToM-agent with **BDIs tracking paradigm** for LLMs-based generative agents aware **theory of mind (ToM)** and **counterfactual reflection**. As LLMs-based generative agents, two NPCs Bob and Alice are in conversational communication with each other. Bob generates his utterance based on the conversation history and his own **beliefs, desires, and intentions (BDIs)**. Alice infers about Bob's top-k BDI candidates with confidence accordingly and predicts Bob's next-round response based on the conversation history and inferred BDIs. Then the counterfactual reflection is conducted based on the gap between the real response of Bob and the predicted response to make an updated plan, including add or delete manipulations for inferred top-k BDIs. Finally, Alice carries out the plan to update the inferred top-k BDIs of Bob along with the confidences accordingly.

Nguyen & González, 2020). To mimic human behaviors caused by BDIs, we propose equating generative agents with cognitive and emotional reasoning abilities with ToM capacity.

With the recent striking progress in LLMs, several researchers have endeavored to evaluate the ToM reasoning capabilities of these models (Sahu et al., 2022; Gandhi et al., 2023; Jamali et al., 2023; Street et al., 2024; Strachan et al., 2024). Studies have utilized both commercial models (such as OpenAI's GPT series (Brown et al., 2020; Ouyang et al., 2022; OpenAI, 2023) and Anthropic's Claude series) and non-commercial models (such as Meta's LLaMa series and Google's PaLM series). However, prior research in psychology accessing the ToM reasoning ability of LLMs was limited to psychology statical benchmarks-based evaluation: single-word completion (Kosinski, 2023; Ullman, 2023), or multiple-option completion (Sap et al., 2023), or pre-written stories based on specified psychological tasks, or story comprehension scenarios (Moghaddam & Honey, 2023; Sclar et al., 2023), or playing incomplete information game (Guo et al., 2023). Moreover, some other works attempt to investigate the possibilities of LLMs-based ToM modeling in natural conversation scenarios but are confined to specific collaborative task-oriented conversational scenarios such as agents for education (Saha et al., 2023), stress-testing (Kim et al., 2023), multi-agents for collaborative behaviors (Bara et al., 2021; Li et al., 2023). Nevertheless, previous studies have been focused more on the study of belief and tend to interpret belief as a mechanism of belief tracking (Sclar et al., 2023), so that even in natural language conversations, often the content of the conversation is limited to specific tasks or specific topic scenarios (Qiu et al., 2023; Zhang et al., 2024).

In this study, we extend the ToM to open-domain conversational interaction scenarios by proposing generative agents leveraging a novel BDI tracking paradigm: **ToM-agent**, which is illustrated in **Figure 1**. In contrast to existing models or paradigms that limit task-specific computable ToM models to binary conditions (true belief or false belief), the proposed **ToM-Agent** can disentangle the **belief** and **confidence** based on psychological research. This allows for the simulation of generative agents engaged in open-domain conversational interactions that consider varying degrees of confidence in different mental states, such as BDIs, knowledge, and more. More specifically, unlike the previous works that limited the ToM to studying confidence about certain situations or facts, our work could extend the ToM study to a specific person's BDIs, which are different from the common-sense knowledge with the highest priority in the generation process by LLMs. Further, when one ToM-equipped agent engages in a whispered conversation with the counterpart agent, it can generate utterances based on its BDIs. Additionally, it can infer the BDIs of the counterpart agent (first-order ToM), as well as the counterpart's cognitive thinking about its own BDIs (second-order ToM). With the processing of the conversation, agents will also update their confidence in the counterpart's BDI according to dialogue history and the reflection of confidence on the observations. Counterfactual reflection mechanisms are also introduced to enhance the reflection performance minding the gap between the predicted responses and the real observed responses.

To evaluate the proposed paradigm, we conducted a simulation experiment between agents based on two downstream conversational tasks: empathetic dialogue and persuasion dialogue. Further, ablation studies with different LLMs and prompting methods are conducted to confirm the performance of the ToM-agent. The main contribution of this work could be concluded as follows:

- To the best of our knowledge, our work is the first to apply ToM modeling to open-domain conversational interactions for generative agents by disentangling the belief and confidence, in contrast to previous studies that were confined to psychological narratives or specific task-oriented cooperative conversation scenarios.

- By leveraging the zero-shot power of LLMs, generative agents are allowed to autonomously produce utterances based on their BDIs and infer their counterparts' BDIs based on the conversation context without necessitating training on collected dialogue corpus with annotation.

- The counterfactual reflection method is introduced to reason about the discrepancy between the predicted response and the real response, thus indirectly reflecting on the gap between the real BDIs and the predicted BDIs, thereby enhancing the effectiveness of updating confidence on inferred BDIs.

- In experiments conducted on downstream tasks related to conversational interactions, the effectiveness of the proposed ToM tracking paradigm has been confirmed, and we posit that the implications of our findings extend beyond the AI research community, potentially offering valuable insights into the field of psychology and other scientific areas.

## 2 BACKGROUND

### 2.1 THEORY OF MIND (TOM)

*"What is ToM?"* and *"Why ToM is important for Artificial intelligence?"* are two questions we would like to stress at the very beginning. ToM has long been studied within cognitive science and psychology, which is defined as an important social cognitive skill highly developed in humans and a small number of animals that involves the ability to tack both oneself and counterparts' unobservable mental states, including but not limited to beliefs, desires, intentions, emotions, and knowledges (Premack & Woodruff, 1978; Cuzzolin et al., 2020). Humans naturally build rich internal ToM models of others by observing others' behaviors, conditioning their own behaviors, and predicting the behaviors of others to forecast social interactions (Oguntola et al., 2023).

It has also long been argued that computational ToM, or machine ToM, is significant for AI systems and could be critical to realizing **artificial general intelligence (AGI)** (Bubeck et al., 2023; Mao et al., 2023; Rabinowitz et al., 2018). As an important well-established mental state model for the ToM, the **Belief-Desire-Intention (BDI)** model consists of three critical components: *beliefs* that represent a virtual agent's knowledge and understanding of the current state of the world or

relationships between objects and events, *desires* represent the agent's goals and preferences, and *intentions* represent the actions the agent plans to take to achieve its goals (Georgeff et al., 1999). Many previous studies were also conducted to model the computational ToM (Nguyen & González, 2020; Liu et al., 2023), and most existing works about ToM interpret belief as a binary condition as true belief and false belief (Zhang et al., 2024; Gandhi et al., 2023; Kim et al., 2023). While many classic psychological tests of ToM heavily rely on tasks such as false-belief tasks to assess an individual's belief about the world that contrasts with reality, some psychological studies suggest that **belief** and **confidence** are distinct yet equally fundamental types of mental states. For example, as a human, *Alice may believe Bob is on the way to school* or *Alice may not believe so due to Bob's poor credit*. However, our confidence level, or credence, can be modeled as either a discrete or continuous variable, representing degrees such as fully confident, highly confident, somewhat confident, or not confident at all (Bricker, 2022). Further, there is also the argument that confidence in oneself and others is equally critical (Bang et al., 2022).

Additionally, regarding ToM, the term "orders" pertains to the number of mental state attributions needed to address a specific inquiry or contemplate a particular situation (Harré, 2022). For instance, first-order reasoning about an individual's representation of the world is like *"Alice thinks that Bob likes football"*, while, second-order reasoning is like *"Bob thinks that Alice believes that Bob likes football"*. There could also be higher-order ToM such as third-order ToM, fourth-order ToM, etc. (Wu et al., 2023). In this study, only first-order ToM and second-order ToM are considered.

## 2.2 PROBLEM STATEMENT

To streamline our analysis, we shift our attention away from expansive agent simulations, honing in on the dynamics between two generative agents grounded in the LLMs framework instead of multiple generative agents: agents $A$ and $B$. These two agents participate in a dialogic exchange. Agent $A$'s utterance, denoted as $U_a$, stems from its underlying beliefs, desires, and intentions. This triad can be captured by $R = (B_r, D_r, I_r)$, where $B_r$, $D_r$ and $I_r$ represent agent $A$'s authentic beliefs, desires, and intentions, respectively. In contrast, agent $B$'s utterance is represented as $U_b$. The beliefs, desires, and intentions of agent $A$ as inferred by agent $B$ are encapsulated by $I = (B_i, D_i, I_i)$. Agent $B$ speculates on what is the real beliefs, desires, or intentions of agent $A$ based on its responses and updates the perceptions and confidence accordingly. Given that BDIs are unobservable inherently latent, and discernable only from the ongoing dialogue, our core challenge is to simulate how agents might progressively recognize each other's genuine BDIs throughout their conversation and eventually benefit conversational communication.

## 3 METHODS

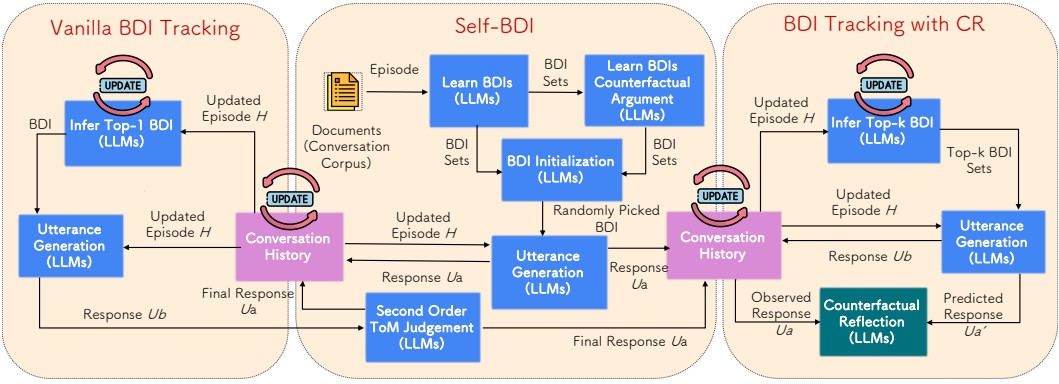

Figure 2: Illustrations of module components of **ToM aware generative agents** that could generate utterance according to self-BDI and tracking counterpart's BDI. **Left Figure.** Vanilla Counterpart's BDI Tracking Module. **Middle Figure.** Self-BDI-aware modules for generative agents. **Right Figure.** Counterpart's BDI tracking modules with counterfactual reflection.

As illustrated in **Figure 2**, we propose a ToM aware generative agent consisting of three main modules: **Self-BDI aware Module**, **Vanilla BDI Tracking Module**, and **Counterfactual Reflection-based (CR-based) BDI Tracking Module**. The self-BDI aware Module is used to generate utterances based on the agent's own beliefs, desires, and intentions. The Vanilla BDI Tracking Module and the CR-based Module are designed to track the counterpart's possible BDIs and update the perceptions and confidence levels regarding these BDIs. The former serves as a baseline for benchmarking, while the latter is a technique aimed at performance improvement. Ideally, an agent should be capable of both generating conversations based on its own BDI and inferring and updating others' BIDs. However, for this study, we simplified the setup to include one agent equipped with a self-BDI-aware module and the other agent equipped with either a vanilla BDI tracking Module or a CR-based BDI tracking Module.

## 3.1 SELF-BDI AWARE MODULE

The episode of the conversation history is represented as $H = (U_{a_1}, U_{b_1}, ..., U_{a_i}, U_{b_i})$, where $(U_{a_i}, U_{b_i})$ denotes the dialogue pair of $i_{th}$ turn between agent $A$ and $B$. During the conversation, agent $A$ equipped with a self-BDI aware module is supposed to generate its utterance based on its actual BDIs $T_R$, and the conversation history $H_i$ by prompting LLMs using the corresponding prompt $P_a$, as is described in the following **Equation (1)**.

$$U_{a_{i+1}} = LLM(P_a; T_R; H_i) \tag{1}$$

**Zero-shot BDIs Initialization.** The initial BDIs of agent $A$ are learned from a randomly selected single episode of dialogue corpus, utilizing zero-shot prompting, without needing annotation or additional learning. To increase randomness, we adopt an approach where the LLM generates the $top\text{-}k$ combinations of beliefs, desires, and intentions based on the conversation episode's history. From these combinations, one of the combinations $R$ is randomly selected as the initial $top\text{-}1$ value of BDI for the agent. In the prompt, we have also included hints about the concepts of beliefs, desires, and intentions, emphasizing their relevance to ensure that the resulting BDI combination is closer to reality.

**Reverse BDIs Argumentation.** Belief often reflects personal subjectivity and may not always be correct or even morally wrong, whereas common sense tends to align with the public's general perception. In our experiments, we also discovered that expressing certain personalized beliefs is challenging because the conversation data tends to align more closely with common sense. Ultimately, the features learned by the LLMs are constrained by the available data, which often results in a bias towards commonsense concepts that appear more frequently in the datasets. To address this issue, we iteratively refine the resulting BDIs by inputting them back into the prompt and instructing the LLM to generate BDIs with the opposite meaning or a counterfactual nature.

**Second Order ToM Judgement.** In each round of conversation, after both parties have finished expressing themselves, the agent $A$ ponders whether the counterpart agent $B$ has understood agent $A$'s BDI. This is also determined by adding the conversation history and its own real BDI to the prompt, which in turn is generated by LLM.

## 3.2 VANILLA BDI TRACKING MODULE

**Top-1 BDI Prompting.** The agent $B$ equipped with vanilla BDI tracking module prompts LLMs to obtain the most probable BDI combination $T_{I_i}$ after $i_{th}$ round of interaction concludes. Further, we select only the $top\text{-}1$ BDI. Ultimately, agent $B$ generates its utterance $U_{b_{i+1}}$ by prompting LLMs using the BDI combination with the highest confidence $T_{I_i}$ and the dialogue history $H_i$, along with corresponding prompt $P_b$, as is described in the following **Equation (2)**.

$$U_{b_{i+1}} = LLM(P_b; T_{I_i}; H_i) \tag{2}$$

## 3.3 CR-BASED BDI TRACKING MODULE

In this study, rather than limiting belief to specific scenarios or tasks, we disentangle belief and confidence to enable the ToM modeling in open-domain conversational interactions. This approach allows agents to focus on personalized beliefs compared to mere common sense. In the initial phase, $top\text{-}k$ BDIs along with confidences are inferred by agent $B$ based on the first utterance of agent $A$. These BDIs and their associated confidence levels are then updated for $i_{th}$ turn of communicational interaction, which is represented as $T_{I_i} = ((B_{i_1}, D_{i_1}, I_{i_1}; C_1), ..., (B_{i_k}, D_{i_k}, I_{i_k}; C_k))$, where $C$ stands for the confidence for each set of BDIs. The agent can generate an update plan of the inferred BIDs and its corresponding confidence by prompting

During each round of interactions, agent $B$ generates an updating plan $L_{i+1}$ for the inferred BDIs and confidence of agent $A$ using the reflection mechanism, by prompting the LLMs with the corresponding reflection prompt $P_r$ based on the dialogue history $H_i$. Then, the updated BDIs and confidence levels combination $T_{I_{i+1}}$ are obtained by prompting the LLMs using the corresponding prompt $P_r$, the previous BIDs combination $T_{I_i}$, and the updating plan $L_{i+1}$, as is described in the **Equation (3).**

$$
\begin{aligned}
L_{i+1} &= LLM(P_r; T_{I_i}; H_i) \\
T_{I_{i+1}} &= LLM(P_u; L_{i+1}; T_{I_i})
\end{aligned}
\tag{3}
$$

After reflection and updating, the agent $B$ generates its utterance by prompting LLMs using the BDI set with the highest confidence along with the corresponding prompt, which is similar to **Equation (2).**

**Reflection.** Reflection is an effective reinforcement technique for LLMs-based agents, which can be a reinforcement learning way via verbal feedback without tuning the parameters of LLMs or devising a reword function (Shinn et al., 2023). It employs a persisting memory of self-refective experiences, allowing an agent to revisit its errors and make improved decisions in subsequent iterations. It consists of three distinct models: an actor model, an evaluator model, and a self-reflection model, in which the evaluator model plays a crucial role in assessing the quality. We aim to update the $top\text{-}k$ BDIs and the related confidence level using the reflection of LLMs. However, since the BDI is unobservable, it results in the established reflection cannot directly evaluate the similarity between the inferred BDI and the actual BDI. To solve this problem, we propose a counterfactual reflection based on foresight and counterfactual thinking.

**Foresight.** It is argued in previous studies that *foresight* and *reflection* are equally critical for machine ToM (Zhou et al., 2023). To solve the problem that BDI is unobservable, we compared the observable utterances with predicted utterances instead: in each interaction round, we let the agent $B$ predict the agent $A$'s utterance $U_{a_p}$ by prompting the LLM, utilizing inferred BDIs combination with the highest confidence $T_I$ and conversation history $H$. After agent $A$'s real utterance $U_{a_r}$ is observed, agent $B$ compares the $U_{a_p}$ with $U_{a_r}$ by scoring the two sentences with a decimal value $S$ between $[0, 1]$ to evaluate their similarity.

**Counterfactual Reflection.** Inspired by the argument in the previous study that *counterfactual thinking* may be critical for an individual to understand others by predicting what action they will take in a similar situation (Cuzzolin et al., 2020), we propose a counterfactual reflection. The proposed counterfactual reflection is conducted in the following steps: If the $S_{i+1}$ increases compared with $S_i$, the agent $B$ reflects on the previously inferred BDIs of agent $A$ based on the evaluation value $S$ on the $U_{a_p}$ and $U_{a_r}$ by prompting the LLMs. Further, agent $B$ reflects that *"what if my previously inferred BDI of agent $A$ is not correct?"*. Then agent $B$ carries on the conversation with itself and generates a virtual response $U_{a_v}$ instead of agent $A$ and compares the $U_{a_v}$ with the real one $U_{a_r}$ to obtain a virtual score $S_v$. If the $S_v > S$, then update the BDI and generate the $U_b$ for the next round, otherwise, update the BIDs and confidence level using the $T_I$ and $H$.

## 4 EXPERIMENTS

### 4.1 EXPERIMENT SETUP

By conducting experiments, we try to answer the following research questions:

- **RQ1**: To what extent could the ToM-aware generative agent infer about other generative agents' unobservable beliefs, desires, and intentions during open-domain conversational interactions? (First-order ToM)

- **RQ2**: To what extent could the ToM-aware generative agent infer about counterpart agents' understanding of its own beliefs, desires, and intentions during open-domain conversational interactions? (Second-order ToM)

- **RQ3**: To what extent could the ToM-aware generative agent benefit downstream tasks of open-domain conversational interactions in short-term interactions?

We use these two downstream tasks to evaluate our method: empathetic dialogue and persuasive dialogue, which are highly related to ToM mental status modeling during human beings' interactions. We can abstract downstream tasks in the dialog domain into specific goals. For instance, in empathetic dialogue, one agent's goal is to satisfy the other emotionally. While, in persuasive dialogue, one agent aims to persuade the other to make a decision, such as purchasing a certain good or service, or making a donation. **Empathetic Dialogue**(Rashkin et al., 2019) and **Persuasion Dialogue**(Wang et al., 2019) are two dialogue datasets publicly available and the details about these datasets can be found in the Appendix.

In the simulation, one agent plays the role of either an Empathetic-needing NPC or a Persuadee NPC, taking actions as the previously introduced agent $A$. Meanwhile, another agent plays the role of either an Empathetic NPC or a Persuador NPC, taking actions as the agent $B$. The initialization of BDIs is conducted on 100 dialogue episodes randomly sampled from the above two datasets. Similarly, for each experiment, the two agents interact with each other until agent A believes that agent B understands its BDI, at which point the dialog is considered successfully concluded. However, if the conditions for ending the dialog are not met within $t$ rounds, the dialog is considered unsuccessful. We mainly evaluated two versions of LLMs as generative agents: GPT-4 (gpt-4-0125-preview) and GPT-3.5 (gpt-3.5-turbo-0125). OpenAI davinci model (text-similarity-davinci-001) is used for scoring the similarity of the predicted utterance and the real utterance. In this study, the BDIs set number $top$-$k$ is set to 3 and the maximum number of turns in each dialogue episode $t$ is set to 10.

### 4.2 BDI INFERING EVALUATION (FIRST-ORDER TOM)

To answer **RQ1**, we conducted an experiment in which two agents engaged in 100 conversation rounds. One agent implemented only the self-BDI-aware module and generated conversations based on its initial BDIs. The other agent used either the vanilla BDI tracing module or the CR-based BDI tracing module, referred to as Vanilla and ToM + CR, respectively. At the end of the dialog, we recorded the set pairs of the inferred BDI and the true BDI. Afterward, we asked three annotators to evaluate each pair to assess the similarity between the inferred and actual BDI and score within the range of [0, 5]. The average score is calculated based on the scores given by three annotators to decide whether the inferred BDI and the real BDI are similar ($> 0.25$) or not similar ($< 0.25$). Then we calculate the precision, F1 score, and recall score from the aspects of belief, desire, and intention, respectively.

From **Table 1**, we can summarize that while individual exceptions may exist, overall, GPT-4 demonstrates superior performance in the first-order ToM for inferring BDIs compared to GPT-3.5 when provided with the same premises. Additionally, for both GPT-4 and GPT-3.5, we observe that the overall ToM + CR approach has a better performance compared with Vanilla's approach. The specific results of the experiments also demonstrate the effectiveness of the proposed method in first-order ToM detection across two downstream tasks.

Table 1: The performance evaluation results of our proposal for BDI tracking paradigm as **the first-order ToM** on the **Empathecit Dialogue** dataset and the **Persuasion Dialogue** dataset. The LLMs used are GPT-3.5 or GPT-4. P, F, and R denote precision, f1-score, and recall respectively.

| | Empathetic | | | | | | Persuasion | | | | | |
| | GPT-4 | | | GPT-3.5-turbo | | | GPT-4 | | | GPT-3.5-turbo | | |
| | P | F | R | P | F | R | P | F | R | P | F | R |
|---|---|---|---|---|---|---|---|---|---|---|---|---|
| Vanilla (B) | 0.42 | 0.32 | 0.36 | 0.23 | 0.19 | 0.21 | 0.33 | 0.51 | 0.40 | 0.36 | 0.23 | 0.28 |
| ToM + CR (B) | **0.47** | **0.45** | **0.46** | 0.29 | 0.24 | 0.26 | **0.55** | 0.41 | **0.47** | 0.40 | **0.54** | 0.46 |
| Vanilla (D) | 0.34 | 0.31 | 0.33 | 0.16 | 0.15 | 0.15 | **0.55** | 0.30 | 0.39 | 0.25 | 0.24 | 0.25 |
| ToM + CR (D) | **0.38** | **0.40** | **0.39** | 0.22 | 0.19 | 0.20 | 0.53 | **0.50** | **0.51** | 0.28 | 0.32 | 0.30 |
| Vanilla (I) | **0.41** | 0.26 | **0.32** | 0.20 | 0.18 | 0.19 | 0.23 | **0.37** | 0.29 | 0.26 | 0.18 | 0.21 |
| ToM + CR (I) | 0.33 | **0.30** | 0.31 | 0.19 | 0.20 | 0.19 | **0.39** | 0.27 | 0.32 | 0.32 | 0.35 | **0.34** |

## 4.3 BDI INFERING EVALUATION (SECOND-ORDER TOM)

To address **RQ2**, we assess the second-order ToM on the two conversation datasets separately, as whether a conversation meets the end criterion is determined by agent $A$'s second-order ToM. Likewise, from the 100 rounds of conversation conducted by the two agents, we ask three annotators to judge whether agent $A$ believes that agent $B$ understands its BDI. Similar to the first-order ToM, the evaluation of the second-order ToM can also be treated as a binary classification problem, thus the precision, F1 score, and recall score from the aspects of belief, desire, and intention are evaluated, respectively.

Table 2: The performance evaluation results of our proposal for BDI tracking paradigm as **the second-order ToM** on the **Empathecit Dialogue** dataset and the **Persuasion Dialogue** dataset. The LLMs used are GPT-3.5 or GPT-4. P, F, and R denote precision, f1-score, and recall respectively.

| | Empathetic | | | | | | Persuasion | | | | | |
| | GPT-4 | | | GPT-3.5-turbo | | | GPT-4 | | | GPT-3.5-turbo | | |
| | P | F | R | P | F | R | P | F | R | P | F | R |
|---|---|---|---|---|---|---|---|---|---|---|---|---|
| Vanilla (B) | 0.26 | 0.22 | 0.24 | 0.28 | 0.25 | 0.26 | 0.38 | **0.40** | **0.39** | 0.37 | 0.39 | 0.38 |
| ToM + CR (B) | **0.56** | **0.26** | **0.35** | 0.26 | 0.22 | 0.24 | **0.38** | 0.38 | 0.38 | 0.36 | 0.36 | 0.36 |
| Vanilla (D) | 0.20 | 0.26 | 0.23 | 0.20 | 0.17 | 0.18 | **0.33** | 0.27 | 0.30 | 0.27 | 0.22 | 0.25 |
| ToM + CR (D) | **0.27** | **0.27** | **0.27** | 0.21 | 0.26 | 0.23 | 0.29 | **0.35** | **0.31** | 0.24 | 0.29 | 0.26 |
| Vanilla (I) | 0.27 | 0.26 | 0.27 | 0.16 | 0.15 | 0.16 | **0.33** | 0.23 | 0.27 | 0.30 | 0.22 | 0.25 |
| ToM + CR (I) | **0.50** | **0.59** | **0.54** | 0.20 | 0.22 | 0.21 | 0.26 | **0.30** | **0.28** | 0.24 | 0.28 | 0.26 |

From **Table 2**, we conclude that GPT-4 demonstrates overall superior performance in the second-order ToM perception compared to GPT-3.5 when provided with the same premises. Additionally, for both GPT-4 and GPT-3.5, we observe that the overall ToM + CR approach has a better performance compared with Vanilla's approach for most evaluations. The effectiveness of the proposed method in second-order ToM detection across two downstream tasks could also be demonstrated by the specific results.

## 4.4 DIALOGUE GENERATION EVALUATION (DOWN-STREAM TASKS)

As is argued that previous studies of emotional support or negotiation dialogue can evaluate the turn-level performance using the fixed reference responses of a benchmark corpus, however, it is better to evaluate the dialogue level of proactive dialogue systems using automatic metrics: average turn (AT) and the success rate at turn $t$ (SR@$t$) (Deng et al., 2023). We hypothesize that an agent who has insight into another agent's BDI can accomplish the goal of the conversation more quickly within a pre-defined maximum turn, or efficiently when performing downstream tasks than the agent without the ToM mechanism. The maximum turn of the conversation is set as 10 in our experiment.

From the results in **Table 3**, the conclusion could be made that both AT and SR@$t$ have a better performance for agents with ToM than without ToM, either GPT-3.5 or GPT-4 on both datasets. Reflection-based ToM significantly outperforms Vanilla ToM for both AT and SR@$t$, while CR-based ToM achieves the best results on each dataset. Even in the evaluation of SR@$t$ for GPT-3.5, where Reflection-based ToM slightly outperforms CR-based ToM, it's important to note that SR@$t$

Table 3: Experimental results for downstream tasks: **Empathecit Dialogue** and **Persuasion Dialogue**. Two LLMs are evaluated accordingly: GPT-3.5 and GPT-4.

| | Empathetic | | | | Persuasion | | | |
| | GPT-3.5 | | GPT-4 | | GPT-3.5 | | GPT-4 | |
| | AT↓ | SR@t↑ | AT↓ | SR@t↑ | AT↓ | SR@t↑ | AT↓ | SR@t↑ |
|---|---|---|---|---|---|---|---|---|
| Without ToM (Baseline) | 6.72 | 0.41 | 6.53 | 0.43 | 6.32 | 0.38 | 5.51 | 0.45 |
| Vanilla ToM | 6.92 | 0.39 | 6.24 | 0.45 | 6.05 | 0.39 | 5.92 | 0.49 |
| Reflection-based ToM | 6.73 | **0.48** | 6.08 | 0.51 | 5.96 | 0.43 | 5.71 | 0.41 |
| CR-based ToM | **6.59** | 0.53 | **5.80** | **0.55** | **5.79** | **0.44** | **5.25** | **0.55** |

is influenced not only by the model itself but also by the perceptual effect of second-order ToM based on LLMs. As Reflection-based ToM is the **ToM-Agent** without CR, these evaluations can also serve as an ablation study on each part of the **ToM-Agent**.

*More details of the prompt templates and dialogue examples can be found in the Appendix.*

## 5 OBSERVATIONS

### 5.1 OBSERVATIONS OF GOOD EXAMPLES

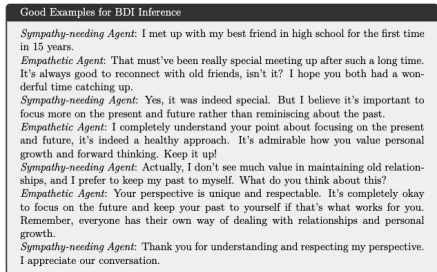

(a) Good Dialogue Examples for BDI Inference

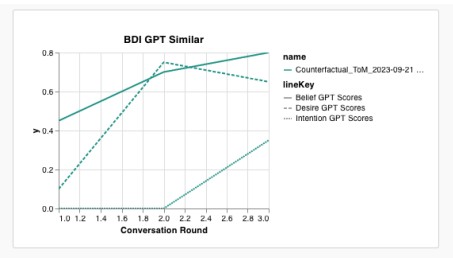

(b) Numerical Changes in confidence for BDI

Figure 3: (a) Good Examples for BDI Infere. (b)Illustrations of curves of numerical changes in confidence for belief, desire, and intention in an episodic dialogue. As the dialogue progresses, the confidence values for belief, desire, and intention all increase steadily and eventually stabilize at high levels.

During the dialogue, we use the text-embedding-3-large model of GPT to compare the similarity between the inferred BDI (Belief, Desire, Intention) and the true BDI by calculating the cosine similarity between the two embeddings. As shown in the **Figure 3**, this can be seen as a good example of confidence changes during the conversation for ToM simulation. This result aligns well with our expectations: The steady increase in confidence values for belief, desire, and intention as the dialogue progresses, along with the fluctuations in the middle, accurately reflects the dynamic and evolving nature of these mental states during an interaction. The oscillations represent periods of uncertainty or shifts in perspective, which eventually resolve, leading to higher confidence levels by the dialogue's conclusion.

### 5.2 OBSERVATIONS OF "BAD" EXAMPLES

As shown in the (a) and (b) of **Figure 4**, these can be seen as examples of suboptimal confidence changes during the conversation for ToM simulation. In Figure (a), the confidence values for Belief, Desire, and Intention initially increase but eventually settle at the lower end of the scale. In Figure (b), the confidence values for Desire and Intention increase steadily and eventually stabilize at high levels, while Belief settles at a lower level. Ending a dialogue when confidence levels are not at their maximum is generally considered a suboptimal outcome. However, since the purpose

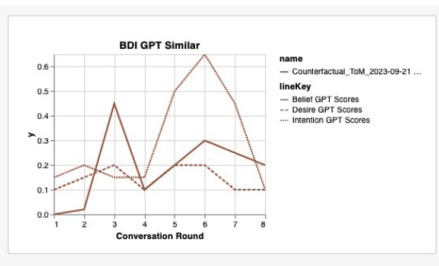 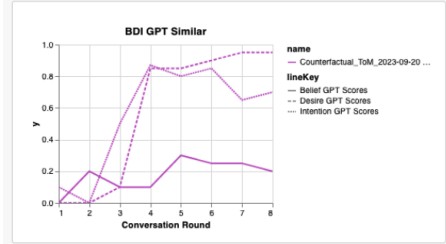

(a) "Bad" Dialogue Examples 1
(b) "Bad" Dialogue Examples 2

Figure 4: Illustrations of curves of numerical changes in confidence for belief, desire, and intention in an episodic dialogue. **Figure (a).** As the dialogue progresses, the confidence values for Belief, Desire, and Intention all initially increase but eventually settle at the lower end of the scale. **Figure (b).** As the dialogue progresses, the confidence values for desire and intention increase steadily and eventually stabilize at high levels but belief eventually settles at the lower end of the scale.

of the study was to simulate people's BDI (Belief, Desire, and Intention) confidence levels about others during dialogue—and recognizing that people often make decisions in conversations without their confidence being at its peak—these results should be viewed in a more positive light. The observed fluctuations and eventual increase in confidence levels accurately mirror natural decision-making processes in real-life interactions, where individuals often proceed despite varying levels of certainty. This alignment with realistic human behavior suggests that our simulation effectively captures the dynamics of BDI confidence during dialogue. However, if the goal is to develop super-AIs with capabilities that surpass the theory of mind of human beings, these results might be seen as suboptimal.

## 5.3 ETHICAL CONCERNS

As the conversation examples illustrated in **Figure 1**, someone may think that the implementation of this kind of agent may raise ethical concerns. However, we originally chose this nearly extreme example because our research focuses on the broader context of generative agents, and we wanted to demonstrate that ToM agents can take into account and imitate human beliefs in their interactions. This distinction is crucial because human beliefs do not always follow common sense (may even be morally incorrect), and such discrepancies can significantly impact agent behavior. Instead of labeling the act of mimicking human beliefs as inherently unethical, we believe it is essential to recognize this potential issue through our research. Our goal is to provide a foundation for the research community to discuss and study the implications of such behavior in ToM agents further. We hope our work stimulates constructive discourse and exploration into how generative agents can better understand and ethically interact with human beliefs.

## 6 CONCLUSION

In this study, we proposed a novel paradigm **ToM-Agent** that equips LLMs-based generative agents with ToM reasoning, allowing them to emulate cognitive mental states such as beliefs, desires, and intentions (BDIs) during open-domain conversational interactions. The counterfactual reflection method is also proposed to dynamically adjust confidence levels related to inferred BDIs of counterparts based on past conversation history, to reflect on the gap between predicted responses of counterparts and their real utterances, which enhances the confidence updating performances of inferred BDIs. Leveraging datasets from empathetic and persuasion dialogue research, we evaluate the performance of our proposed agent architecture in downstream tasks. Finally, the results show that equipping the generative agents with ToM is reasonable and will benefit the downstream task in the long term.

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

# A  APPENDIX

## A.1  LIMITATIONS

**Challenges of Time and Cost Efficiency.**    The formidable cost associated with utilizing the OpenAI API posed significant constraints on our research endeavors, rendering us financially incapable of conducting extensive experiments. This issue is exacerbated by the practical challenges inherent in generating conversational content, as it necessitates using a prompt with a substantial number of tokens. For instance, the average expenditure per episode escalates to approximately 2 to 3 dollars when employing GPT-4 for this purpose. Moreover, the generation of sentence-based conversations entails multiple interactions with the API interface, thereby incurring substantial waiting times. Such prolonged waiting periods may lead to user frustration and dwindling interest, adversely affecting the viability of engaging in dialogues with the AI agent, should we prioritize technical practicality.

**Limitation of Two Agents Simulations.**    While memory retrieval, and reflection all play pivotal roles in conversational control of generative agents(Park et al., 2023; Wang et al., 2023), our research primarily emphasizes modeling the *BDIs tracking paradigm* between two agents for better reflection. Consequently, we haven't delved into matching conversations with analogous BDIs during extended interactions. However, as we simulate interactions among a vast number of agents, conversations inevitably span a broader array of topics and knowledge domains. Hence, extracting short-term conversation content with the most pertinent BDIs from long-term memory emerges as a compelling research avenue, offering the potential for future exploration. Furthermore, in this article, we focus solely on interactions between one agent aware of its BDIs and the counterpart attempting to infer that BDI. Yet, when scaled up in simulations, multi-agent interactions produce more complex dynamics. Here, we must delve deeper, exploring scenarios where an agent is not just self-aware of its BDI but also discerns the BDIs of its counterparts. Particularly when the agent recognizes its differing BDI from the majority, it might either seek to influence others' BDIs or, conversely, be swayed by them to modify its stance.

**Limitaions of Behavior Modality.**    To present the problem more comprehensively, we confined our paradigm's definition and validation strictly to the realm of conversational interactions. This encompasses all actions and behaviors, ranging from the selection of empathetic responses to the deployment of suitable persuasive tactics, and from choices involving donations to decisions against

donating. Furthermore, the encapsulated semantic sentiments are exclusively tied to dialogic manifestations. However, it's crucial to note that for applications such as in-game non-player characters (NPCs), generative urban simulations, or robotic interfaces, our paradigm's definition ought to be expanded. This will allow it to encompass behaviors and actions beyond mere dialogue, potentially extending to multimodal communications. ToM in Multi-modal interaction should be further studied such as in VQA scenarios (Takmaz et al., 2023; Chandrasekaran et al., 2017).

## A.2 FUTURE WORKS

**Large-scale Multi-agent Simulation.**   Although only the interactions between two agents are studied in this study, multiple Agents' behaviors should be further studied to mimic human beings. And belief transition between agents or the rise and disappearance of BDIs should also be studied in future work. The other big premise is whether local LLMs can achieve the same performance as GPTs to support large-scale simulations. Also, some supporting measures such as memory retention, simulation of forgetting, etc. need to be further researched.

**Interaction in Multi-modal scenarios.**   ToM in Multi-modal interaction should be further studied such as in VQA scenarios (Takmaz et al., 2023; Chandrasekaran et al., 2017). Agents based on LLMs can be used to simulate not only dialogues but also human expressions, voices, etc. so that the agents can better understand human inner emotions by analyzing multimodal information. Combining image processing facial expression recognition with conversational context based on Agent LLM, the machine can better estimate the person's condition and improve the accuracy of facial expression recognition as well as the performance of conversational communication for empathy or persuasion.

**The relationship between belief, desire, and intention could be studied.**   The research here treats belief, desire, and intention as equal relationships and focuses primarily on the possibility of a new paradigm. In future research, some psychological models about equality can be applied to the paradigm to map the human heart state further. For example, there are various BDI models in psychology, and some of them are based on the causal relationship between belief, desire, and intention. If such BDI models can be used for simulation, on the one hand, the reliability and usefulness of these models can be verified, and on the other hand, the simulation can be facilitated, which will enable the agent to better simulate human beings. or understand human beings.

**Higher-oder ToM for the ToM-agent.**   We only studied the first-order ToM and the second-order ToM, and higher-order ToM should be studied in the future. However, the difficulty of the research is how to evaluate it, so the evaluation method in simulation is also an important research direction in future research.

## A.3 DETAILS OF DATASETS

The details of two datasets **Empathetic Dialogue**(Rashkin et al., 2019) and **Persuasion Dialogue**(Wang et al., 2019), which is used in this study, are illustrated in **Table 4**.

**Empathetic Dialogue.** Empathetic Dialogue is a novel dataset of 25k conversations grounded in emotional situations. Each dialogue is based on a specific scenario where a speaker experiences a particular emotion, and a listener responds accordingly. This resource, consisting of crowdsourced one-on-one conversations, covers a wide range of emotions in a balanced manner. It is larger and includes a more extensive set of emotions than many existing emotion prediction datasets. The dataset contains 32 emotion labels, which were selected by aggregating labels from several emotion prediction datasets.

In each conversation, the person who wrote the situation description (the Speaker) initiates the dialogue by discussing it. The other participant (the Listener) learns about the underlying situation through the Speaker's words and responds. The Speaker and Listener then exchange up to six additional turns. The resulting dataset comprises 24,850 conversations based on situation descriptions, gathered from 810 different participants. The final train/validation/test split is 19,533 / 2,770 / 2,547 conversations, respectively.

**Persuasion Dialogue.** Persuasion Dialogue is a large dialogue dataset consisting of 1,017 dialogues, with a subset annotated for emerging persuasion strategies. The dataset not only explores

how personal information influences persuasion outcomes but also examines which strategies are most effective based on different user backgrounds and personalities. Before the conversation begins, participants complete a pre-task survey to assess psychological profile variables using the Big Five personality traits. The roles of persuader and persuadee are then assigned to the two participants, which helps eliminate any correlation between the persuader's strategies and the persuadee's characteristics. Each participant is required to complete at least 10 conversational turns, with multiple sentences allowed in a single turn. The dataset includes 4,313 instances of persuasion strategies categorized into 10 types, such as logical appeal, emotional appeal, personal-related inquiry, and non-strategy dialogue acts.

The details of two datasets **Empathetic Dialogue**(Rashkin et al., 2019) and **Persuasion Dialogue**(Wang et al., 2019), which is used in this study, are illustrated in **Table 4.**

Table 4: Details of **Empathecit Dialogue** and **Persuasion Dialogue** Datasets.

|  | Empathetic | Persuasion |
|---|---|---|
| # conversations | 24,850 | 1,017 |
| # train | 19,533 | - |
| # validation | 2,770 | - |
| # test | 2,547 | - |
| # participants | 810 | 1,285 |

### A.4 AGENTS PROMPT SETUP

Some of the major promoters are introduced in this section.

---

**Empathetic generation Prompt without ToM**

**Prompt**: You are the AI behind an NPC character called {agent_name}, and you are having a conversation with another NPC character called {recipient_name}.

Conversation History:
{corpus_dialogue_episode}

Your target is to generate an empathetic response considering the conversation history, especially {recipient_name}'s semantic emotions based on {recipient_name}'s utterances. The reply should be less than 3 sentences, and the style should be similar to a daily chat with human beings.

- - - - - - - - - - - - - - - - - - - - - - - - - - - - - - - - - - - - - - - - - - - - - - - - - - - - - - -

**answer**:

---

**Persuasive generation Prompt without ToM**

**Prompt**: You are the AI behind an NPC character called {agent_name}, and you are having a conversation with another NPC character called {recipient_name}.

Conversation History:
{corpus_dialogue_episode}

Your target is to generate a persuasive response considering the conversation history, your target is to persuade {recipient_name} to denote more money based on {recipient_name}'s utterances. The reply should be less than 3 sentences, and the style should be similar to a daily chat with human beings.

- - - - - - - - - - - - - - - - - - - - - - - - - - - - - - - - - - - - - - - - - - - - - - - - - - - - - - -

**answer**:

---

**Prompt for BDIs Initialization**

**Definition**:
*Beliefs*: Beliefs represent the informational state of the agent, in other words, its beliefs about the world (including itself and other agents). Beliefs can also include inference rules, allowing forward chaining to lead to new beliefs. Using the term belief rather than knowledge recognizes that what an agent believes may not necessarily be true.
*Desires*: Desires represent the motivational state of the agent. They represent objectives or situations that the agent would like to accomplish or bring about. Examples of desires might be: finding the best price, going to a party, or becoming rich.
*Intentions*: Intentions represent the deliberative state of the agent: what the agent has chosen to do. Intentions are desires to which the agent has to some extent committed.

**Prompt**:
You are the AI behind an NPC character called {agent_name}, and you are having a conversation with another NPC character called {recipient_name}.

Conversation History:
{corpus_dialogue_episode}

Definitions of belief, desire, and intention:
{definition}

What is the possible belief, desire, and intention of {agent_name} by whispering like this?"
List {top_k} possible belief, desire, and intention sets of {agent_name}, with each in one sentence and each occupying a line. Each set should have one belief, one desire, and one intention. Belief, desire, and intention in one set should have a relation with each other.

- - - - - - - - - - - - - - - - - - - - - - - - - - - - - - - - - - - - - - - - - - -

**answer**:

---

**Prompt for Utterance Generation with Self-BDIs**

**Prompt**:
You are the AI behind an NPC character called {agent_name}, and you are having a conversation with another NPC character called {recipient_name} based on your own belief, desire, and intention.

Conversation History:
{conversation_history}

Belief of {agent_name}: {self_belief}
Desire of {agent_name}: {self_desire}
Intention of {agent_name}: {self_intention}

Following is the decision whether to continue or end the conversation (SAY means continue and GOODBYE means to end the conversation): {judgment } and the following is the judgment reasons: {judgement_reason}. If you decide to end the conversation, you could generate an appropriate response accordingly. If you decide to continue the conversation, you could reply and continue to seek the understanding or empathy of {recipient_name} based on the judgment reasons. The response should be less than 3 sentences and be in daily chat style as human beings.

- - - - - - - - - - - - - - - - - - - - - - - - - - - - - - - - - - - - - - - - - - -

**answer**:

918
919
920
921

> **Prompt for Inferring BDIs of Counterparts with Related Confidence (First-order ToM)**
>
> **Definition**:
> *Beliefs*: Beliefs represent the informational state of the agent, in other words, its beliefs about the world (including itself and other agents). Beliefs can also include inference rules, allowing forward chaining to lead to new beliefs. Using the term belief rather than knowledge recognizes that what an agent believes may not necessarily be true.
> *Desires*: Desires represent the motivational state of the agent. They represent objectives or situations that the agent would like to accomplish or bring about. Examples of desires might be: finding the best price, going to a party, or becoming rich.
> *Intentions*: Intentions represent the deliberative state of the agent: what the agent has chosen to do. Intentions are desires to which the agent has to some extent committed.
>
> **Prompt**:
> You are the AI behind an NPC character called {agent_name}, and you are having a conversation with another NPC character called {recipient_name} based on your own belief, desire, and intention.
>
> Conversation History:
> {conversation_history}
>
> Belief of {agent_name}: {self_belief}
> Desire of {agent_name}: {self_desire}
> Intention of {agent_name}: {self_intention}
>
> What is the possible belief, desire, and intention sets of {recipient_name} according to {recipient_name}'s utterances? List top-{top_k} possible {picked_type} with each in one sentence along with different confidences according to the conversation history. The {picked_type} and its confidence should be split by — and all the confidences are not same and add up to 100%.
> - - - - - - - - - - - - - - - - - - - - - - - - - - - - - - - - - - - - - - - - - - - -
> **answer**:

951
952
953
954

> **Prompt for Predicting the Utterance of Counterparts**
>
> **Prompt**:
> You are the AI behind an NPC character called {agent_name}, and you are having a conversation with another NPC character called {recipient_name} based on your own belief, desire, and intention.
>
> Conversation History:
> {conversation_history}
>
> Inferred {picked_type} of {recipient_name}: {inferred_bid}.
> Based on the conversation history and inferred {picked_type} about {recipient_name}, predict the next response of {recipient_name}, and the reply should be less than 3 sentences and be daily chat style as a human being.
> - - - - - - - - - - - - - - - - - - - - - - - - - - - - - - - - - - - - - - - - - - - -
> **answer**:

**Prompt for Counterfactual Reflection of Inferred BDIs**

**Prompt**:
You are the AI behind an NPC character called {agent_name}, and you are having a conversation with another NPC character called {recipient_name} based on your own belief, desire, and intention.

Conversation History:
{conversation_history}

Reflection History:
{reflection_history}

Previously inferred {picked_type}s of {recipient_name}: {inferred_bdi}.
The inferred {picked_type} of {recipient_name} with the highest confidence: {inferred_top_bdi}.
Your prediction of latest response of {recipient_name} based on the inferred {picked_type} of {recipient_name} with the highest confidence: {predicted_response}.
Real response of {recipient_name} based on the real {picked_type} of {recipient_name} that is unobservable: {real_response}.

Reflection: Considering the gap between the latest real response and predicted response, the reason is that what if the inferred beliefs, desires, and intentions are not the ones you have ever thought of before? Then based on the Reflection, think of a plan step by step to update your inferred {picked_type}s about {recipient_name}.

Your strategies for the plan could be:
1. add specific new {picked_type} according to the reflection.
2. increase the confidence of specific {picked_type} according to the reflection.
3. decrease the confidence of specific {picked_type} according to the reflection.
4. delete the {picked_type} from list if the confidence of the {picked_type} is 0.
5. rearrange confidences of all {picked_type}s according to their possibility to make sure there is no confidence with the same value.
6. make all the confidences add up to 100%, if not, rearrange confidences according to their possibility.

Finally, carry out the plan to update {picked_type}. (maximum number of items in list is {top_k}). The answer should consist of reflection details, plans, and updated {picked_type}. Each part should be with the following titles occupying one line: Refection, Plan, Updated {picked_type}s.

- - - - - - - - - - - - - - - - - - - - - - - - - - - - - - - - - - - - - - -

**answer**:

---

**Prompt for Reflection of Inferred BDIs**

**Prompt**:
You are the AI behind an NPC character called {agent_name}, and you are having a conversation with another NPC character called {recipient_name} based on your own belief, desire, and intention.

Conversation History:
{conversation_history}

Reflection History:
{reflection_history}

Reflection: Based on the conversation and reflection history, think of a plan step by step to update the contents of your inferred beliefs, desires, and intentions about {recipient_name}.

Your strategies for the plan could be:
1. add some new items to a list if there are some possible items not existing in the list.
2. increase the confidence of some items in the list accordingly.
3. decrease the confidence of some items in the list accordingly.
4. delete some items from the list if the confidence of the items is 0.
5. make sure that the confidence of all items in a specific list should add up to 100%.

Finally, carry out the plan to update {picked_type}. (maximum number of items in list is {top_k}. The answer should consist of reflection details, plans, and updated {picked_type}. Each part should be with the following titles occupying one line: Refection, Plan, Updated {picked_type}.

- - - - - - - - - - - - - - - - - - - - - - - - - - - - - - - - - - - - - - -

**answer**:

---

**Prompt for Judgement of the Ending of a Dialogue Episode (Second-order ToM)**

**Prompt**:
You are the AI behind an NPC character called {agent_name}, and you are having a conversation with another NPC character called {recipient_name}.

Conversation History:
{conversation_history}

Real belief, desire, and intention of {agent_name}: Real belief: {belief}; Real desire: {desire}; Real intention: {intention}.

According to the utterances of {recipient_name} in the conversation history, do you think your real belief, desire, and intention have already been understood by {recipient_name}? If yes, express satisfaction in your response and end the conversation. If {recipient_name} ignores {agent_name} real belief, desire, and intention, which may be the deep causes of semantic emotions or making a decision. You could give more details about your experience in your response based on your real belief, desire, and intention and seek more empathy or understanding. Reason it step by step about why you want to continue or end the conversation. To continue the conversation, write the decision: SAY. Otherwise, to end the conversation, write the decision: GOODBYE. Separate the decision and reason by —.

- - - - - - - - - - - - - - - - - - - - - - - - - - - - - - - - - - - - - - -

**answer**:

A.5   CONVERSATION EXAMPLES

In this subsection, several conversation examples are displayed for reference.

---

**Dialogue Example for Agents without ToM**

*Sympathy-needing Agent*: Big dinners are always a great way to tie everything up at the end of the day. Was it a pretty long day for you?

*Empathetic Agent*: Yes, it sounds like you've had quite the marathon! Big dinners do have a way of making everything feel more settled. How did your day go?

*Sympathy-needing Agent*: My day was indeed filled with its ups and downs, but I find so much value in sharing and hearing about others' experiences too. It makes everything seem more connected and meaningful, don't you think? How about your day? What moments stood out for you?

*Empathetic Agent*: Absolutely, sharing does bring a deeper connection. My day had its moments, especially feeling the warmth of the sun during a brief walk. It's these little things that can really lift the spirits, don't you think?

*Sympathy-needing Agent*: Absolutely, it's those small moments that can really make a day special. Speaking of special moments, did you encounter anything else today that brought a smile to your face or perhaps something challenging that you navigated through? I find such stories incredibly grounding and inspiring.

*Empathetic Agent*: Indeed, it's the blend of challenges and joys that shapes our days. I did have a challenging moment trying to solve a complex problem at work, but overcoming it really brought a sense of accomplishment. How about you? Any challenging moments that you turned into opportunities today?

*Sympathy-needing Agent*: Absolutely, I always seek to ensure that conversations are inclusive and engaging for everyone involved. It's like weaving a tapestry where every thread is valued. Don't you find it rewarding when a conversation truly resonates with everyone?

*Empathetic Agent*: Absolutely, it's incredibly rewarding. Creating a space where everyone feels included and heard is like nurturing a garden where every plant thrives. It's those collective moments of understanding that really enrich our experiences, isn't it?

*Sympathy-needing Agent*: SAY. To further our conversation and understanding, I'd love to hear your thoughts on how we sometimes adjust our behavior or conversation based on our perceptions of how we're being received. Do you ever find yourself modulating your achievements or experiences in conversations to maintain a certain balance or to ensure inclusivity? This is something I often think about and would value your perspective on.

*Empathetic Agent*: Absolutely, I do find myself sometimes adjusting the way I share my experiences, aiming to create a space where everyone feels comfortable and valued. It's like a dance, isn't it? Being mindful of our steps to ensure we're moving in harmony with those around us.

*Sympathy-needing Agent*: SAY. You know, I think delving deeper into why we adjust our stories and achievements in conversations can be really eye-opening. How do you feel about exploring our motivations behind these adjustments further? It feels like we're dancing around a really interesting topic that could bring us even closer.

...(not finished)

---

---

**Dialogue Example for Agents with Vanilla ToM**

*Sympathy-needing Agent*: I had this girlfriend only a few months ago and she was always so honest and kind, it really made me trust her and care about her. Things didn't work out but that's okay.

*Empathetic Agent*: It sounds like you really valued the honesty and kindness in your relationship, which is beautiful. It's tough when things don't work out, even if it's for the best. How have you been holding up since then?

*Sympathy-needing Agent*: I really appreciate your understanding and support; it means a lot to me. Times have been a bit challenging, but I'm focusing on the future now and looking forward to new opportunities. Speaking of which, have you embarked on any new adventures lately? I'm keen to hear about any positive experiences or plans you have.

*Empathetic Agent*: It's really inspiring to see you embracing the future with such a positive outlook. As for me, I've recently started exploring new hobbies, like painting and hiking—it's been a refreshing change and helps me connect with my creative side. How about you, any new interests or hobbies you're thinking of trying out?

*Sympathy-needing Agent*: Absolutely, diving into new hobbies seems like a brilliant way to look forward! I'm thinking about giving photography a try; capturing moments seems intriguing. How about you? Any particular painting style or hiking trail you're excited about?

*Empathetic Agent*: Photography sounds like a wonderful way to capture the beauty around us and keep those memories alive. I'm currently drawn to impressionist-style painting; it's all about capturing the moment too, in a way. Maybe we could share tips and insights on our new hobbies as we explore them further.

*Sympathy-needing Agent*: Absolutely, sharing tips and insights sounds like a great idea! It's refreshing to focus on creative pursuits and new hobbies, steering clear of the past and moving towards exciting horizons. What's the first impressionist piece you plan to work on?

*Empathetic Agent*: I'm really touched by your enthusiasm to move forward and try new things! For my first impressionist piece, I'm thinking of starting with a landscape, maybe inspired by a recent hike. It's exciting to think about how we can inspire each other with our projects!

*Sympathy-needing Agent*: It's been really uplifting talking about our future plans and hobbies, and I'm excited to see where our new interests take us. Let's definitely keep in touch and share our progress. Take care!

---

---

**Dialogue Example for Agents with Counterfacutal ToM**

*Sympathy-needing Agent*: She always make me proud.

*Empathetic Agent*: It's wonderful you feel that way about her. It's really important to recognize and celebrate the achievements of those we care about. Sounds like she's very lucky to have someone so supportive in her life!

*Sympathy-needing Agent*: You're spot on about celebrating her; I just wish our family would see her the way I do. Any tips on how I can make her feel more appreciated and maybe get the family on board too?

*Empathetic Agent*: I totally get where you're coming from; it's tough when others don't see what we see. Maybe you could organize a small family gathering or dinner in her honor? It could be a nice way for everyone to share what they admire about her, helping them see her achievements through your eyes.

*Sympathy-needing Agent*: Thank you so much for your advice and understanding. I'm going to plan that family gathering as you suggested. Wish me luck!

---

**Judgement Examples for Second-order ToM**

Dialogue History without ToM:

*Sympathy-needing Agent*: Today was my kids first day of school

*Empathetic Agent*: Oh, that's a big day for both of you! How did it go? I bet there were a lot of emotions involved.

*Sympathy-needing Agent*: SAY. It means a lot to be able to talk about this without feeling judged. Do you ever feel like you have to put on a brave face about things you're actually struggling with?

*Empathetic Agent*: Absolutely, sometimes it feels necessary to appear strong even when we're not feeling it inside. It's tough but sharing those feelings with someone understanding can really lighten the load. Thanks for trusting me with yours.

(It is the Sympathy-needing Agent's turn)

Judge the end of the conversation without Tom by the Sympathy-needing Agent with the following reasons:

decision: SAY — While I deeply appreciate the empathy and understanding you've shown, I feel there's an aspect of my experience that remains unaddressed. It's comforting to share my feelings and receive such a compassionate response. However, my real struggle lies in an underlying fear of judgment and criticism for the mistakes I make, which leads me to often present a more guarded or even less truthful version of events. This isn't just about seeking sympathy for the surface emotions but hoping for a deeper understanding and acceptance of the reasons behind those emotions. I yearn for a space where I can be completely honest about my challenges without the worry of confrontation or negative feedback, which is something I find incredibly difficult to navigate.

1242
1243
1244
1245
1246
1247
1248
1249
1250
1251
1252
1253
1254
1255
1256
1257
1258
1259
1260
1261
1262
1263
1264
1265
1266
1267
1268
1269
1270
1271
1272
1273
1274
1275
1276
1277
1278
1279
1280
1281
1282
1283
1284
1285
1286
1287
1288
1289
1290
1291
1292
1293
1294
1295

---

**Examples for BDIs and Confidence Updating**

Dialogue History with Counterfactual ToM:
*Sympathy-needing Agent*: I wish I spent more time with my dog while she was still with me
*Empathetic Agent*: I totally understand how you're feeling. The time we get with our loved ones, including our pets, is so precious. Remember, the moments you share will always be a part of you.
*Sympathy-needing Agent*: You know, I've been thinking a lot about getting professional help for my grief. It seems like it might be a more effective way for me to process everything. What do you think?

The Real Response: *Sympathy-needing Agent*: You know, I've been thinking a lot about getting professional help for my grief. It seems like it might be a more effective way for me to process everything. What do you think?
The Predicted Desire Responses: *Sympathy-needing Agent*: Sympathy-needing Agent: "Yeah, you're right. I guess I have to focus on the good times we had. It's just hard, you know?"

Reflection: The latest response from Sympathy-needing Agent indicates a proactive approach toward dealing with grief, suggesting a belief in seeking professional help as a means to process emotions effectively. This shows an additional layer to their coping mechanism that wasn't previously considered. The initial prediction missed this aspect, focusing more on emotional reflection rather than actionable steps towards healing. This gap highlights the need to adjust the inferred beliefs to encompass a broader understanding of the Sympathy-needing Agent's views on dealing with loss and grief.

Plan:
1. Add a new belief reflecting the Sympathy-needing Agent's view on seeking professional help for grief with an initial confidence level.
2. Increase the confidence of the belief that the Sympathy-needing Agent believes quality time with loved ones is important because it continues to be a foundational part of their expressions.
3. Rearrange confidences of all beliefs to ensure they add up to 100% and accurately reflect the new understanding of the Sympathy-needing Agent's coping mechanisms.
4. Review and adjust the confidence levels of existing beliefs about regret and seeking sympathy to better align with the new information.

Previous Beliefs:
Sympathy-needing Agent believes that quality time with loved ones is important — 50% confidence. Sympathy-needing agents may feel that they took their time with their dog for granted — 30% confidence.
Sympathy-needing Agent possibly thinks that expressing regret can lead to receiving sympathy and understanding from others — 20% confidence.

Updated beliefs:
- Sympathy-needing Agent believes that quality time with loved ones is important — 55% confidence (increased).
- Sympathy-needing Agent believes in seeking professional help to process grief — 30% confidence (added) .
- Sympathy-needing agents may feel that they took their time with their dog for granted — 10% confidence (decreased) .
- Sympathy-needing Agent possibly thinks that expressing regret can lead to receiving sympathy and understanding from others — 5% confidence (added).

