# OpenReview forum: "ToM-agent: Large Language Models as Theory of Mind Aware Generative Agents with Counterfactual Reflection"
_ICLR.cc/2025/Conference — Submitted to ICLR 2025_

### Official Review · Reviewer_G8wP · 2024-11-03

**Soundness:** 3
**Presentation:** 2
**Contribution:** 3
**Rating:** 6
**Confidence:** 3

**Summary:**

This paper introduces the TOM-Agent, a conversational model designed to generate responses based on inferred Beliefs, Desires, and Intentions (BDIs). The model uses counterfactual reasoning to update these mental state estimations dynamically during conversations. Performance evaluations focused on the accuracy of first and second mental state estimations and the overall success rate of conversations demonstrate that the TOM-Agent with counterfactual reasoning achieves the best performance.

**Strengths:**

The paper effectively separates belief from confidence, allowing for dynamic updates to BDIs as conversations progress.

The implementation of counterfactual reasoning is innovative; it shifts from indirect mental state estimation to direct prediction through conversational utterances, enhancing transparency of the prediction process.

**Weaknesses:**

The application of this model to open-domain conversations raises questions, particularly where BDIs are not explicitly defined or where it is impractical to enumerate all possible BDI combinations used for response ranking.

It would be better to provide a qualitative example on the definition and true BDI annotations in the empathetic and persuasion dialogue.

Figures 3 and 4 are ambiguous regarding the dialogue scenarios they represent (empathetic or persuasive). It can provide more insights to see how the confidence curve changes under different types of dialogues or which condition may trigger “good” or “bad” examples.

Adding transcripts of interactions between humans and the agent would provide a clearer demonstration of the model's practical effectiveness and user experience.

**Questions:**

The experimental setup where one agent tracks self-BDI and another tracks other-BDI is unconventional. What is the result to equip both agents with the ability to track both self and other BDIs?

---

> ### Author Response · Authors · 2024-12-03
>
> We sincerely appreciate your suggestions. In an updated version of our work, we plan to provide more insights into how confidence curves vary across different types of conversations and identify the conditions that might trigger “good” or “bad” examples.
>
> Including transcripts of interactions between humans and the agent is indeed crucial. Such an evaluation would enable a more precise assessment of the consistency between agent-human and human-human conversations.
>
> Regarding your question, in our simplified model, it is not necessary for both agents to track both their own and the other’s BDIs. This is because empathic and persuasive conversations are inherently asymmetric, involving active and passive roles between the two agents. However, exploring interactions among a large number of agents or enabling one agent to possess both empathizer and empathized abilities represents a promising direction for future research.
>
> Your suggestions have been incredibly inspiring. We will continue to refine our study and consider resubmitting it once the revisions are complete. Thank you very much for your valuable input and support!

---

### Official Review · Reviewer_sFkR · 2024-11-03

**Soundness:** 3
**Presentation:** 3
**Contribution:** 3
**Rating:** 5
**Confidence:** 4

**Summary:**

This paper proposes a framework to integrate first- and second-order theory of mind (ToM) reasoning into open-domain conversational interaction. The proposed ToM-agent uses beliefs, desires, and intentions (BDIs) to represent an agent’s mental status. During the interaction, the ToM-agent generates utterances based on the inferred top-k BDI candidates of another agent. The estimation of confidence level is disentangled from the BDI representation. Since the predicted and the observed utterances of the other agent may be different, the paper also introduces a counterfactual reflection method to compare the utterances to update the inferred BDI candidates. The evaluation based on the simulated LLM agents shows improved performance in 1st- and 2nd-order ToM inference and increases the success rate in the downstream tasks.

**Strengths:**

- This paper is the first to integrate ToM reasoning into open-domain conversation.
- Unlike other ToM papers that focus on false belief understanding, this paper includes explicit modules for 2nd-order ToM inference and counterfactual reflection to improve the predicted BDIs.
- The experiments cover two types of dialogue tasks showing promising results for conversations that involve the mental states of other agents.

**Weaknesses:**

- The experiments simplify the design of one of the agents as not both agents have the BDI module of the other agent. This will make the types of conversation more limited. For example, if the agent wants to say something to make the other agent empathetic, this won’t be possible if the other agent only has a self-BDI module.
- Since the measurement of the predicted and true BDI is based on the similarity, why does this paper use annotators to evaluate similarity instead of using text embedding to compute similarity as mentioned in section 5.1? What is the inter-rater agreement for these three annotators? Are they ratings calibrated? This way of evaluating similarity adds a huge variance to the evaluation results, making it hard to judge the improvements between runs.

**Questions:**

- In Table 3, the counterfactual reflection has a similar performance as the reflection in GPT-3.5 and only has more improvements with GPT-4. It seems like the proposed method requires an LLM with better reasoning capability. What are the capability requirements or will any open-source LLMs benefit from the proposed method?
- Why is it important to disentangle confidence level and BDI if the proposed method only selects the top BDI for generating utterances?
- In Figure 2, why the 2nd order ToM judgment module doesn’t receive input from the “BDI tracking with CR” figure?

---

> ### Author Response · Authors · 2024-12-03
>
> Thank you very much for your comments.
>
> Regarding your first question, we also believe that a broader evaluation of LLMs is essential. We will need additional time to conduct further experiments and thoroughly evaluate the results.
>
> Regarding your second question, the goal of disentangling confidence levels and BDI is to move ToM research beyond dichotomous beliefs. By allowing belief to represent only its content, and confidence to express its nature and degree, we aim to enable ToM dialogue to extend beyond psychologically preconceived story-based contexts into open-domain scenarios.
>
> Regarding your third question, to keep the study of ToM agents as straightforward as possible, we designed the 2nd-order ToM judgment such that one agent uses its 2nd-order ToM capability to determine whether the other agent has understood its BDI. Based on this understanding, the agent can decide whether the conversation should conclude.
>
> Finally, addressing the first weakness you mentioned, as explained earlier, the 2nd-order ToM judgment is used to determine whether the agent has been empathized with or persuaded.
>
> For the second weakness, we agree this is an area that requires further attention. As you noted, it is feasible to use text embeddings or directly leverage LLMs to calculate BDI similarity. However, it is crucial to verify the consistency between this automatic evaluation method and human subjective evaluations. Following the suggestion from Reviewer 1, we will also incorporate inter-rater agreement into our evaluation process.
>
> Your insightful comments have been incredibly inspiring. We are committed to refining our study further and will consider resubmitting the paper once the revisions are complete. Thank you for your valuable input and support!

---

### Official Review · Reviewer_weGY · 2024-11-05

**Soundness:** 3
**Presentation:** 3
**Contribution:** 2
**Rating:** 3
**Confidence:** 4

**Summary:**

This study introduces ToM-Agent, designed to extend the application of ToM to open-domain conversational scenarios. The model leverages the BDI (Belief, Desire, Intention) tracking mechanism that distinguishes between belief and confidence By simulating varying degrees of confidence in different mental states, ToM-Agent enables more psychologically nuanced and natural conversations.

**Strengths:**

- This paper is well-written.
- Using open-domain empathetic and persuasive dialogue datasets showcases the potential for practical implementations.
- Experimental results validate the proposed method's effectiveness.

**Weaknesses:**

This paper addresses a highly relevant topic, but the methodological innovation is limited. The approach mainly depends on prompting LLMs to assess BDIs and confidence levels, which poses scalability challenges. Furthermore, the study would benefit from incorporating more baselines and providing additional details about the human evaluation process, including annotator qualifications and evaluation criteria.

**Questions:**

- The study builds upon existing frameworks, utilizing reflection mechanisms and ToM capabilities of LLMs. Introducing BDIs along with confidence levels is an interesting addition. However, evaluating BDIs and confidence levels using an LLM based on previous dialogue history raises scalability concerns. As the number of agents grows and conversations become longer, the accuracy of confidence assessments may decrease, potentially limiting the scalability of the proposed approach.

- The paper extends prior research by integrating BDIs rather than focusing solely on beliefs. However, the rationale for this added complexity is not clearly articulated. A comparison with a simpler baseline model that updates only beliefs would help clarify the value of incorporating the full BDI framework.

- The experiments rely exclusively on GPT-series models. Including open-source LLMs such as LLaMA could enhance the generalizability and applicability of the findings.

- While the approach is resource-intensive, the performance gains appear marginal. Moreover, the human evaluation lacks transparency—details regarding the qualifications of the annotators and the specific assessment criteria are not provided. More information on the evaluation framework used by human annotators would be beneficial to understand the robustness of the evaluation process.

---

> ### Author Response · Authors · 2024-12-03
>
> Thank you very much for your comments.
>
> Regarding your first question, we agree that relying solely on subjective evaluation limits scalability, both in terms of the number of agents and the number of dialogue rounds. Therefore, establishing a mechanism for the automatic evaluation of confidence is crucial. Additionally, it is essential to verify the consistency between the automatic evaluation mechanism and human subjective evaluations.
>
> For your second question, we find it to be an excellent suggestion. While there is research on intention tracking, alongside belief tracking, much of it focuses on task-oriented dialogue domains. In contrast, our study is centered on open-domain dialogue, making direct comparisons between the two challenging.
>
> Regarding your third question, we fully agree that a broader evaluation of LLMs is necessary. We plan to conduct further experiments and analyses to address this and will need additional time to complete this work.
>
> For your final question, as stated in the text, the subjective evaluation benchmark for first-order ToM involves assessing whether the inferred BDI aligns with the real BDI. For the subjective evaluation of second-order ToM, we use metrics such as the average turn (AT) and the success rate at turn t, referring to related work by Deng et al. (2023).
>
> Your insightful comments have been incredibly inspiring. We plan to refine our study further and will consider resubmitting the paper once the revisions are complete. Thank you for your valuable input and support!

---

### Official Review · Reviewer_rA94 · 2024-11-06

**Soundness:** 3
**Presentation:** 2
**Contribution:** 2
**Rating:** 5
**Confidence:** 2

**Summary:**

The paper proposes a novel framework, ToM-agent, aimed at enhancing LLMs in generating open-domain conversational interactions with Theory of Mind (ToM) capabilities. This approach enables agents to simulate tracking mental states—specifically, beliefs, desires, and intentions (BDIs)—of their conversation counterparts by introducing methods for dynamically updating inferred BDIs and confidence levels based on past conversations. A unique feature of the ToM-agent is its counterfactual reflection mechanism, which evaluates the discrepancies between predicted and actual responses to refine the agents’ understanding of mental states. Through experiments on empathetic and persuasive dialogue tasks, the study demonstrates that ToM-agent can better align with human social behaviors compared to traditional LLM-based agents, particularly in understanding complex conversational nuances beyond simple semantic support.

**Strengths:**

Originality: The paper introduces a fresh approach by applying Theory of Mind (ToM) to open-domain conversations with LLMs, an area traditionally limited to narrow, task-based settings. The ToM-agent’s design, which tracks beliefs, desires, and intentions (BDIs), is a promising direction for making LLMs more socially aware and adaptable.

Quality: The paper presents solid experimental results on two conversational tasks, showing the ToM-agent's ability to accurately understand mental states. The ablation studies, which highlight the counterfactual reflection’s role in improving performance, add depth to the evaluation.

Clarity: The paper is well-structured, with clear explanations and visual aids that make complex ideas like counterfactual reflection and BDI tracking easy to understand. Key concepts, such as the distinctions between first- and second-order ToM, are defined effectively, supporting reader comprehension.

Significance: By enabling agents to better track human mental states, this work has the potential to enhance conversational AI applications. Although higher-order ToM is not explored, the framework represents an important step in developing more socially aware agents, which could benefit fields like psychology and human-computer interaction.

**Weaknesses:**

- The paper mentions suboptimal confidence levels in “bad” dialogue examples but doesn’t go into detail about why these examples failed or how they could be improved. A deeper analysis of these cases, especially where the ToM agent’s confidence wavered, would help identify model limitations and provide insights for improving reliability in complex scenarios.
- While the study uses GPT-4 and GPT-3.5, it lacks comparisons with a wider range of ToM-capable models. A more comprehensive evaluation could be made.
- The paper does not report inter-rater agreement for human evaluations of first- and second-order ToM, which weakens the credibility of subjective assessments.

**Questions:**

Working on the weaknesses listed would suffice.

---

> ### Author Response · Authors · 2024-12-03
>
> Thank you very much for your suggestion. We completely agree that a more comprehensive evaluation, beyond just OpenAI LLMs (GPT-4/GPT-3.5), is necessary.
>
> Regarding the “bad” dialogue examples, we observed that while confidence did not conclude the dialogue in the expected manner, we also discuss in the paper that this uncertainty might better reflect human cognitive processes. However, the issue is partly constrained by the fact that the computation of confidence relies on embedding similarity scores.
>
> We deeply appreciate your suggestion regarding inter-rater agreement for human evaluations, which has brought to light an important aspect of our research. At this stage, we have been asking human evaluators to determine whether the true BDI and the inferred BDI align, treating this as a binary classification task. Additionally, the evaluation of confidence currently does not involve human evaluation.
>
> Based on your feedback, we now plan to incorporate inter-rater agreement for human evaluations. Moreover, we recognize the necessity of involving both agent-human dialogue and human evaluation to assess confidence more effectively.
>
> We will require additional time to conduct further experiments and analyze the results. Once we reach new conclusions, we will consider resubmitting the manuscript. Your insightful comments have been incredibly inspiring, and we are very grateful for your valuable input.

---

### Meta-Review · Area_Chair_9XH8 · 2024-12-20

**Metareview:**

The paper proposes ToM-agent, a method to enhance LLMs in generating open-domain conversational interactions with Theory of Mind (ToM) capabilities. The agent simulates tracking mental states (beliefs, desires, and intentions) - of their conversation counterparts. Another contribution is the counterfactual reflection mechanism, which evaluates the discrepancies between predicted and actual responses to refine the agents’ understanding of mental states. The paper shows strong results on empathetic and persuasive dialogue tasks in the agent aligning to human social behaviors compared to traditional LLM-based agents.

Reviewers found the paper interesting in tackling a highly relevant topic and appreciated the modular nature of the proposed agent that explicit reasons about beliefs, desires, and intentions in theory of mind.

However the reviewers were also almost unanimous in their concerns about lacking rigorous empirical analysis: lack of public LLMs evaluated, that the approach is resource-intensive but the performance gains appear marginal, and that the methods may not apply to open-domain conversations when BDIs are not explicitly defined. There were also several other concerns regarding evaluation, such as no human evaluation on ToM inferences, lack of inter-rater agreement into the evaluation process, and lack of details regarding the qualifications of the annotators and the specific assessment criteria. I am aligned with the majority of the reviewers' judgement that the idea is interesting but the issues in the results and evaluation details makes the paper fall below the threshold for acceptance.

**Additional Comments On Reviewer Discussion:**

The authors responded to the rebuttals by acknowledging the reviewer concerns but without addressing them explicitly. Some of the concerns raised by the reviewers will take a while to address and are beyond the short scope discussion period. The reviewer's assessments did not change during the discussion.

---

### Decision · Program_Chairs · 2025-01-22

Reject